# Isolated Fraction of Gastric-Digested Camel Milk Yogurt with Carao (*Cassia grandis*) Pulp Fortification Enhances the Anti-Inflammatory Properties of HT-29 Human Intestinal Epithelial Cells

**DOI:** 10.3390/ph16071032

**Published:** 2023-07-20

**Authors:** Jhunior Abrahan Marcia, Ricardo S. Aleman, Shirin Kazemzadeh, Víctor Manrique Fernández, Daniel Martín Vertedor, Aryana Kayanush, Ismael Montero Fernández

**Affiliations:** 1Faculty of Technological Sciences, Universidad Nacional de Agricultura, Road to Dulce Nombre de Culmí, Km 215, Barrio El Espino, Catacamas 16201, Honduras; jmarcia@unag.edu.hn; 2Doctorate Program in Food Science, University of Extremadura, Avda. de Elvas, s/n, 06006 Badajoz, Spain; 3School of Nutrition and Food Sciences, Louisiana State University Agricultural Center, Baton Rouge, LA 70803, USA; rsantosaleman@lsu.edu (R.S.A.); karyana@agcenter.isu.edu (A.K.); 4Department of Dairy and Food Science, South Dakota State University, Brookings, SD 57007, USA; shirin.kazemzadehpournaki@sstate.edu; 5Nutrition and Bromatology Area, Department of Animal Production and Food Science, University of Extremadura, Avda. Adolfo Suárez, s/n, 06004 Badajoz, Spain; vimanriqu@alumnos.unex.es; 6Department of Nature Conservation and Protected Areas, Government of Extremadura, 06800 Mérida, Spain; daniel.martin@juntaex.es; 7Department of Chemical Engineering and Physical Chemistry, Area of Chemical Engineering, Faculty of Sciences, University of Extremadura, Avda. de Elvas, s/n, 06006 Badajoz, Spain

**Keywords:** camel milk, carao, yogurt, antioxidant activity, anti-inflammatory activity, consumer perceptions

## Abstract

Functional foods have recently generated a lot of attention among consumers looking for healthy options. Studies have examined yogurt with carao to increase health benefits and probiotic characteristics. It has been determined that carao fruit and camel milk have high phenolic compound and antioxidant activity concentrations. The objective of this study was to examine if carao (0, 1.3, 2.65, and 5.3 g/L) incorporated into yogurt enhances anti-inflammatory stimulus and antioxidant activity and impacts the physio-chemical and sensory properties of camel milk yogurt. HT-29 cells were used as a model of anti-inflammatory response, including cytokine responses of IL-8 and mRNA production of IL-1β and TNF-α in gastric digested isolated fraction. In addition, pH, titratable acidity, *Streptococcus thermophilus* counts and *Lactobacillus bulgaricus* counts of camel yogurts were examined during the fermentation process in 0, 2.5, 5, and 7 h whereas viscosity, syneresis, and radical scavenging assay evaluations were determined at hour 7. Furthermore, a consumer study was performed. Compared to control samples, the incorporation of carao into yogurts did not lead to a significant (ρ > 0.05) difference in the pH. In contrast, titratable acidity (TA), viscosity, syneresis, and antioxidant capacity significantly increased with the inclusion of 2.65 and 5.3 g/L carao, while 5.3 g/L carao significantly (ρ < 0.05) increased the counts of both bacteria. The inflammatory response of IL-8 and the level of mRNA production of IL-1β and TNF-α was significantly (ρ < 0.05) lower with 2.65 and 5.3 g/L carao yogurt compared to control camel yogurt. Sensory attributes were not impacted by the addition of 1.3 and 2.65 g/L carao. Carao could be a possible ingredient to consider when improving the nutrition value of yogurt.

## 1. Introduction

Fermented milk can be defined as dairy derivatives based on skimmed, semi-skimmed or whole milk with specific cultures. Dairy derivatives have biologically active components that have historically been of great importance in the diet of different cultures [1]. In the United States, yogurt had a market value of 7.24 billion U.S. dollars in 2021 [2].

The production of this product with a creamy aroma, mouthfeel, and good gel network structure, which impacts the texture of yogurt, is under research using different milk resources and fortifying materials [3].

Camel milk and fermented camel products are popular among Middle East and African people for their nutritious value, therapeutic properties, better functional value than cow milk, antidiabetic values due to having insulin and insulin-like proteins which activate insulin receptors, as well as high amounts of A2 β-casein and immune-proteins such as lactoferrin, immunoglobulin, and lysozyme [4].

Camel milk comprises 2.9–5.5% fat, 2.4–4.5% protein, 2.9–5.8% lactose, and 0.35–0.9% ash [5]. Camel milk has been reported only for drinking due to lactoferrins prohibiting starter growth [6]. Commercial products of camel milk need to be developed better and need consideration and product development [7].

The protein of camel milk is different from bovine milk; for example, camel milk has heat-stable serum proteins of about 23%, weak desaturated serum proteins bonds with casein, β-casein, and β-lactoglobulin, and very low κ-casein [8,9].

Camel milk does not form curd due to different fermentation processes with lactic acid starters, so the final product has a fragile and poor structure due to the formation of large micelles of casein and small size of fat globules [10].

To prepare good-quality camel milk yogurt, raw materials such as polysaccharides are added to strengthen the gel system [11].

The relationship between reactive oxygen and nitrogen species (which cause oxidative stress) with inflammatory diseases is well known [1,2,3,4,5,6]; therefore, plant extracts that present substances such as flavonoids, polyphenols, and tocopherol with antioxidant capacity have, on many occasions, anti-inflammatory effects and antioxidant properties [12].

Animal studies have indicated that higher concentrations of exogenous antioxidants prevent the type of free radical cancer-related damage [13].

For this reason, it is recommended to have a high uptake of antioxidant dietary supplements that can help lower the risk of developing cancerogenic cells. Asthma, rheumatoid arthritis, inflammation of the intestine (bowel disease), psoriasis, multiple sclerosis, cardiovascular diseases, Alzheimer’s, type 2 diabetes, and cancer are diseases associated with chronic inflammation [14,15]. Plants that contain antioxidant dietary supplements and other trace elements may contribute antioxidant and anti-inflammatory properties [16,17].

In this sense, the carao plant (*Cassia grandis*) could be used to improve the chemical properties of the camel milk yogurt obtained. Carao is one species from 500 species of the *Leguminosae* family [18].

Carao is also known for its pharmaceutical properties due to the presence of lanthanoids with strong laxative and antibacterial effects [19,20].

Furthermore, the seeds of carao have seed gums used in paper, cosmetics, textiles, and pharmaceuticals since they are non-toxic, eco-friendly, and safe. The gums show the importance of the structural and crystallographic features of galactomannan. The basic structure of the legume galactomannans has a chain of (1–4) linked mannopyranosyl units, and single galactose subunits are attached to side chains at O-6 [21]. *C. grandis* extracts showed considerable amounts of alkaloids, flavonoids, and phenols and excellent antioxidant capacity compared to Trolox [22,23].

In the leaves, carao showed substantial amounts of grandisina, kaempferol, quercetin, and flavonol [24]. Besides the phytochemical characteristics, *C. grandis* has great antidiabetic potential due to its trypsin inhibitory effect [25].

In addition, carao has been shown to improve the acid and bile tolerance of *Streptococcus thermophilus* and *Lactobacillus bulgaricus,* and also improves the rheological properties of yogurt and sensorial properties of stuffed olives [25,26,27,28].

Functional ingredients can improve the nutrition of yogurt. Researchers such as Alemán and collaborators in 2023 determined that fortified yogurt with carao exerts hypoglycemia in STZ-induced diabetic Wistar rats and anti-inflammatory activities in caco2 cells [14]. In our preliminary studies, a rejection threshold was detected when 5.3 g/L was used in yogurt.

The influence of carao-fortified camel yogurt fractions on HT 29 cells properties is unknown. As a result, this study aims to examine the antioxidant, anti-inflammatory, and consumer perception characteristics of camel milk yogurt as affected by carao.

## 2. Results and Discussion

### 2.1. Physico-Chemical Evaluations

The results of pH, total acidity, viscosity, and syneresis are shown in Figure 1. The pH values of yogurt samples during the fermentation process are illustrated in Figure 1A. The hour effect was significant (*p* < 0.05), whereas the concentration effect and the concentration * hour interaction effect, were not significant (*p* > 0.05). The pH values decreased significantly (ρ < 0.05) during 0, 2.5, 5, and 7 h of fermentation. TA values of the yogurt are illustrated in Figure 1B. The main effects of the concentration and hour were significant (ρ < 0.05), whereas the interaction effect concentration * hour was not significant (ρ > 0.05). Yogurts with 5.3 g/L carao had significantly (ρ < 0.05) higher TA values than control samples. TA remained stable until 2.5 h and significantly (ρ < 0.05) increased after 5 h during the fermentation.

Researchers Al-Zoreky & Al-Otaibi, since 2015, have indicated that the pH value of this type of milk varies with respect to camel species and climatic conditions [7]. The pH decreased to 4.6 after 7 h of incubation. At the same time, the titratable acidity increased to 1.02% in the highest amount of carao camel yogurt (CY3), which shows higher titratable acidity. However, the pH of the lowest carao addition (CY1 treatment) did not present significant differences between CY0 and between CY2 (medium) and CY3 (highest) at 7 h of incubation. This could be due to the addition of carao powder which presents fiber and carbohydrates which could have improved the function of yogurt stater culture bacteria and the production of lactic acid. On the other hand, the addition of carao to camel milk could cause fermentative inhibition. However, the results showed that this addition enhanced the fermentation of the raw product since high acidity was achieved. Acidity and pH are fermented products’ two most influential quality parameters. The starter culture ferments the lactose into lactic acid, producing high acidity and low pH compared to unfermented milk [14].

Other variables analyzed in the camel milk yogurt were the viscosity (Figure 1C) and the syneresis (Figure 1D). The highest viscosity (6353 cP) and lowest syneresis (58%) were observed for CY3 treatment, which indicated that carao powder effectively increased immobilization of the aqueous phase, to a more viscous product. The lowest viscosity (6210 cP) and highest syneresis (59%) were observed for the control sample, which had no added carao powder. Researchers have indicated that camel milk yogurt is flowing and very soft, hence the need for the use of thickeners and stabilizers [29].

Syneresis is one of the visual features of yogurt due to the accumulation of whey and water in the yogurt gel and impacts consumers’ acceptability. Syneresis is mainly due to the shrinkage of the gel structure, which separates whey [30]. Syneresis is a phenomenon that depends on starters, temperature, stabilizers, and other factors which can influence syneresis after fermentation.

Due to galactomannan and other polysaccharides, syneresis was decreased, and viscosity increased by adding carao powder [31]. For example, adding stevia in reduced-fat yogurt formed larger casein micelle clusters which are probably due to the interaction of milk proteins with sativoside with hydrophobic interactions [32]. The addition of fiber and carbohydrates from banana fiber and banana peel fiber binds to water molecules and interferes with camel milk components, including proteins, resulting in protein networks that can prevent the free movement of water molecules and reduction of syneresis in a higher concentration of banana fiber and banana peel fiber [33].

### 2.2. Microbial Counts

Figure 2A,B present *S. thermophilus* (ST) and *L. bulgaricus* (LB) counts of yogurt samples during 7 h of fermentation. The ST and LB counts increased significantly (ρ < 0.05) during 0, 2.5, 5, and 7 h of fermentation. For both bacteria, the hour effect was significant (ρ < 0.05), whereas the concentration effect and the concentration * hour interaction effect, were not significant (ρ > 0.05). During the elaboration process of the camel milk yogurt during 7 h of incubation, the microbial counts at 0 h was 7.7 log CFU/mL of *S. thermophilus* (Figure 2A) and 6.26 log CFU/mL or *L. bulgaricus* (Figure 2B) which reached values of 9.95 and 8.82 log CFU/mL, respectively. Different studies indicated that plant chemical compounds enhance the probiotic viability in yogurt and fermented products [33,34]. Similar results of the final count range (7 to 8 log/mL) at 6 h were found in fermented camel milk with different bacterial starter cultures [35].

In general, polysaccharides improve the texture and viscosity of fermented camel milk due to increased lactic acid bacteria [36]. For example, adding the cinnamon extract and fruit extract increased the bacterial counts to a maximum on the 14th day after incubation [37]. Researchers such as Medina and collaborators in 2023 reported that carao at 5.3 g/L did not affect the growth of *S. thermophilus* and *L. bulgaricus* [27]. Other researchers such as Paz and collaborators in 2022 reported that carao did not affect the viability in M17 and MRS broth during 16 h of incubation in *S. thermophilus* and *L. bulgaricus*, respectively [26].

### 2.3. Radical Scavenging Ability Evaluations

The results of the radical scavenging ability of DPPH (Figure 3A) and ABTS (Figure 3B) are shown in Figure 3. The lowest DPPH-scavenging ability was observed in the control (22.6%). In comparison, the higher radical-scavenging ability was observed in the CY3 treatment (39.08%), indicating that the presence of carao powder during the yogurt formation increased the antioxidant activities in all treatments due to having alkaloids, coumarins, flavonoids, free amino acids, amines, phenols, tannins, reduced sugars, resins, saponins, and triterpenes [25].

The total antioxidant activity of fermented camel milk and yogurt depends on different factors, including starter culture, type of enzyme, and hydrolysis of proteins [38]. Results exhibit a range of 20–40% scavenging ability, which increased during fermentation [39]. Other studies studied the behavior of bioactive molecules in yogurts to which the juice of certain fruits such as blueberries, aronia, and grapes was added [40]. There was a significant increase in the content of phenolic compounds and antioxidant activity.

The results of the ABTS scavenging ability of antioxidant compounds in an aqueous phase assay are also shown in Figure 3B. The results are similar to those obtained with the DPPH method. Camel milk yogurt showed a high capacity of antioxidative ability due to the production of oligopeptides, peptones, and free amino acids by microbial proteolytic activity. There is a synergistic relation between products of proteolysis and phenols that resulted in having higher antioxidant potential in fermented products, including camel milk yogurt [41]. Reducing the impact of reactive oxygen species is the crucial role of bioactive compounds and higher antioxidant activity of camel milk yogurt [42]. Carao has a high polyphenolic content and plants with high phenolic content usually provide good antioxidant capacity [18].

### 2.4. Anti-Inflammatory Activity

Figure 4 shows the content of IL-8 produced by HT-29 human intestinal epithelial cells after their induction with TNF-α. The highest IL-8 value (93.84%) was observed for the sample with TNF-α induction which was significantly the highest compared to the other treatments. TNF-α upregulated the production of IL-8 leading to cells producing more IL-8 in the control treatment compared to carao-treated samples. Significantly (*p* < 0.05) lower IL-8 secretion was observed in the sample without TNF-α and carao addition. Addition of carao to camel milk in the elaborated yogurt produced a significant reduction of IL-8. The highest reduction of IL-8 was presented in the yogurt with the highest amount of carao powder (CY3). It is suggested that lactic acid bacteria produces metabolites that have inhibition effects against inflammatory markers such as TNF-α [43]. This is an interesting result since it is shown that the intake of camel milk yogurt would have effective anti-inflammatory properties on this type of cell.

### 2.5. mRNA Levels of Interleukin-1β (IL-1β) and Tumor Necrosis Factor-α (TNF-α)

mRNA levels of IL-1β and TNF-α were increased by the induction of inflammatory responses in HT-29 cells (Figure 5). These cells were induced to inflammation by LPS addition. There are two pathways of LPS inflammation induction. LPS generally connects to TLR-4, resulting in the activation of the inflammatory response, which is on the mitogen-activated protein kinase and NF-κB pathway [44]. The second pathway is alternative, which happens when ROS is produced upon activation of NADPH oxidase. Therefore, the results showed that pretreatment of HT-29 cells with different types of camel milk yogurt with carao powder addition decreased the gene expression of both IL-1β and TNF-α cytokines. The highest IL-1β value belongs to the sample with LPS-induction, significantly higher than the other treatments meaning more inflammatory activity when compared control. Less mRNA levels of IL-1β was observed in the sample without LPS induction and carao addition. The same results are observed with the level of TNF-α. Camel milk yogurt with carao addition had a significant reduction of TNF-α level meaning less inflammatory activity when compared to control. The effects of carao-added camel milk yogurt on the reduction of gene expression are not due solely to the presence of carao powder. Camel milk yogurt can control its inflammatory effects since milk and dairy products have been used as raw materials to isolate bioactive peptides with different health properties [45]. Many of these bioactive peptides are found in the bioactive form in the amino acid sequence of the original protein. They are converted into physiologically active peptides during gastrointestinal digestion and lactic fermentation [46]. Works carried out by Dharmisthaken and collaborators in 2021 show that fermented camel milk has low molecular weight proteins whose peptides could reduce pro-inflammatory cytokines by lipopolysaccharide-treated murine [47]. On the other hand, Shukla and collaborators in 2022 found that *Lacticaseibacillus paraceei* (M11) obtained from fermented camel milk effectively suppressed LPS-induced pro-inflammatory cytokines and their measures such as NO, TNF-α, IL-β, and IL-6i n RAW 264.7 cells [48].

### 2.6. Sensory Evaluation of Camel Milk Yogurt

A descriptive sensory evaluation of camel milk yogurt with carao addition at different concentrations was studied (Table 1). In general, the values of the evaluated attributes were very close from one treatment to another. However, some significant differences were detected for carao addition for each attribute evaluated by consumers. “Appearance”, “Color”, “Consistency”, “Aroma”, “Flavor”, and “Overall Liking” were the attributes with lower scores for the 5.3 g/L carao camel milk yogurt when compared to control yogurt samples. This is an expected result, as carao in high concentrations provides an unpleasant aroma and flavor and can negatively affect sensory perception [28].

The ranking and purchase intent of the carao yogurts is shown in Table 2. After the beneficial statement was given, there was a significant (ρ < 0.05) increase in purchase intent from their original values among 1.3 g/L and 2.65 g/L carao yogurts. Regardless of the beneficial statement provided to the consumers, the purchase intent of 5.3 g/L carao yogurt was significantly (ρ ≥ 0.05) lower than the other yogurt samples. The 2.65 g/L and 5.3 g/L carao yogurt samples had a significantly (*p* ≥ 0.05) higher rank sum value (183, 180, respectively) compared to the control yogurt (0 g/L) (rank-sum = 160). Figure 6 shows the yes responses concerning the acceptability of the yogurt containing carao over the control yogurt. Generally, the carao fortification into yogurt decreased the selection of the product. The probit regression suggests that the maximum concentration of carao in yogurt was 2.3 g/L without affecting the consumers’ choices compared to control yogurt.

Interestingly, the tasters showed a greater purchase preference for camel milk yogurt with carao addition when they were previously informed that the food had significant health properties such as antioxidant and anti-inflammatory activity (Table 2). More and more consumers are more aware and concerned about maintaining a healthy lifestyle, which is reflected in the results found in this work. Subsequently, the consumers were asked to rank the different camel milk yogurt elaborated in the questionnaire. The results show a baseline preference was established in which the CY0 and CY1 treatments were found above 60% of the product acceptability (Figure 6). Only the treatment with the highest dose of carao was below the significant rejection threshold. This result is also interesting since 60% of the consumers responded affirmatively to the possibility of buying the product [49].

## 3. Materials and Methods

### 3.1. Milk and Plant Material

Camel milk was obtained from a local family farms with flash-pasteurized and dried (Drome Dairy, Centennial, CO, USA). *Cassia grandis* fruits were collected from Catacamas Municipality, Olancho Department (Catacamas, Honduras). The pulp, seeds, and shells were separated and grounded with a Retsch SM 100 knife mill (Retsch GmbH, Haan, Germany) (501–700 mm). The pulp of the fruit was blended with water (10% wt/wt) using a magnetic agitator (Bimarloga Científica, 78HW-1, Buenos Aires, Argentina) at 500 rpm at 60 °C for 30 min. Subsequently, the blend was freeze-dried (Virtis Advantage Pro, SP Scientific, Bucks County, PA, USA) at −75 °C and 0.3 Pa for 48 h. The freeze-dried powder was grounded, sealed, and packed to carry out the various analyses.

### 3.2. Yogurt Preparation

We mixed 555 g of the camel milk powder (Drome Dairy, Centennial, CO, USA) for 3 to 5 min with 4 L of distilled water with a commercial immersion blender (Waring Commercial, McConnellsburg, PA, USA) [29]. Carao freeze-dried powder was added to camel milk at four different concentrations: (i) 0.00 g∙L^−1^ (CY0); (ii) 1.30 g∙L^−1^ (CY1); (iii) 2.65 g∙L^−1^ (CY2); and (iv) 5.30 g∙L^−1^ (CY3). The rehydrated camel milk with carao addition was stirred for 10 min at 10 °C and pasteurized at 83 °C for 30 min. After pasteurization, the product was mixed vigorously with the commercial immersion blender for 3 min at 80 °C. Later, the camel milk with carao addition was inoculated and mixed at 41 °C with 6 mL/4 L of camel milk of *Lactobacillus bulgaricus* LB-12 (Chr. Hansen, Milwaukee, WI, USA) and *Streptococcus thermophilus* ST-M5 (1:1 ratio). The inoculated milk was placed into 355 mL Reynolds RDC212-Del-Pak Combo-Pak containers (Alcoa, Inc., Pittsburgh, PA, USA) and incubated at 41 °C to reach a pH of 4.55–4.57. The final product was stored at 4 °C [14,27]. All the experimental treatments were made in three different batches.

### 3.3. Analysis

#### 3.3.1. Physico-Chemical Evaluations

Syneresis was determined by a centrifugal assay in which 25 g of yogurt was centrifuged in an AccuSpin™ 400 centrifuge (Fisher Scientific Instruments, Pittsburgh, PA, USA) at 2000× *g* for 20 min. pH was measured by using a Thermo Orion 3-Star Benchtop pH Meter (Fisher Scientific, Pittsburgh, PA, USA) [32]. Titratable acidity was examined by titrating 9 mL of camel milk yogurt with 0.1 N NaOH and 0.5 mL of phenolphthalein as an acid-base indicator [10]. Apparent viscosity was recorded using a viscometer (Brookfield Engineering Lab Inc., Stoughton, MA, USA) with a helipad stand using a T-C spindle at 20 rpm at 4 °C. The whey separation was quantified and calculated (Equation (1)):(1)$$\mathrm{Syneresis}\;(\%)=(\frac{whey\;\left(mL\right)}{sample\;intial\;weight\;\left(g\right)})\;\times \;100$$

#### 3.3.2. Microbial Analysis

*S. thermophilus* and *L. bulgaricus* counts were determined at incubation time at 0, 2.5, 5, and 7 h during yogurt fermentation. Yogurt was serially diluted with 99 mL of sterilized peptone (0.1% wt/v) and pour-plated in duplicate. *S. thermophilus* was enumerated with *S. thermophilus* agar adjusted to pH 6.8 with 1 N HCl and incubated at 37 °C for 24 h aerobically, and *L. bulgaricus* was enumerated by using MRS agar adjusted to pH 5.2 with 1 N HCl and incubated at 43 °C for 72 h anaerobically. Colony-forming units were estimated with a colony counter (Leica Inc., Buffalo, NY, USA).

#### 3.3.3. Simulated Gastric Phase Digestion of Camel Milk Yogurt

The different camel milk yogurts were submitted to an in vitro gastric phase digestion. Yogurts were freeze-dried, and 0.2 g of powder was dissolved in 4 mL of HCl (0.15 N) and then mixed with simulated gastric fluid (Chemazone, Edmonton, AB, Canada) and porcine pepsin enzyme (Sigma-Aldrich, St. Louis, MO, USA) solution (mix 1:1) at a 1:50 ratio (*w*/*w*) with sample solution (yogurt with HCL). The yogurt control that does not contain carao went through the same gastric digestion process as the carao sample to guarantee the homogeneity of the process. The obtained solution was adjusted to a pH of 3 using HCl (0.15 N) and was incubated at 37 °C for 2 h with constant shaking. Later, the obtained mixture was heated at 90 °C for 10 min. The solution was centrifuged at 2500× *g* (4 °C) for 15 min. The supernatant was collected and freeze-dried for cell culture examinations and biochemical analysis. Filters between 10 and 50 kDa were used to obtain 10–50 kDa fractions using Amicon^®^ Ultra-15 centrifugal filters (Millipore Corporation, Bedford, MA, USA). After the filtration, phenolic and peptide fractions were isolated from the gastric digestates using Strata™-X-C 33 μm Polymeric Strong Cation column (1 g/12 mL, Phenomenex, Torrance, CA, USA). The freeze-dried digestates were dissolved in 1% formic acid and loaded onto the column. The column was washed with HCl (0.1 N) and eluted with methanol and ammonium hydroxide at a 1:1:1:1 ratio.

#### 3.3.4. Radical Scavenging Assay of the Digested Camel Milk Yogurt

The radical scavenging assay was performed using 2,2′-azino-bis (3-ethylbenzothiazoline-6-sulfonic acid) (ABTS) (Sigma-Aldrich, St. Louis, MO, USA) stock solution, then 14.8 mM of ABTS solution was incorporated into potassium persulfate (1:1, *v*/*v*) (5 mM) solution. This solution was incubated at 25 °C in the dark for 16 h. Later, the solution was added to distilled water until obtaining an absorbance of 734 nm. Then, 20 μL digested camel milk yogurt was diluted to 180 μL ABTS+ solution and incubated at 25 °C in the dark for 15 min. A solution of 180 μL ABTS+ solution and 20 μL distilled water was considered a control [39,49]. ABTS+ scavenging activity was then calculated as follows (Equation (2)):(2)$$\mathrm{ABTS}\;\mathrm{radical}\;\mathrm{scavenging}\;\mathrm{activity}\;(\%)=\left(\frac{Abs\;control-Abs\;sample}{Abs\;control}\right)\;\times \;100$$

The DPPH radical scavenging assay was performed using 2,2-difenil-1-picrilhidracilo (DPPH) (Sigma-Aldrich, St. Louis, MO, USA)) stock solution. We mixed 20 μL of yogurt extract with 180 μL 0.1 mM DPPH solution and incubated this at 25 °C in the dark for 30 min. The absorbance was recorded at 515 nm [50,51]. A solution of 180 μL DPPH+ solution and 20 μL ethanol was considered a control. The DPPH scavenging activity was calculated as shown in Equation (3):(3)$$\mathrm{DPPH}\;\mathrm{Inhibition}\;(\%)=\left(\frac{Abs\;control-Abs\;sample}{Abs\;control}\right)\;\times \;100$$

#### 3.3.5. HT-29 Cell Lines Culture

The HT-29 epithelial cells (human colorectal) were obtained from American Type Culture Collection (ATCC Manassas, VA, USA) to examine anti-inflammatory properties. Cells were cultured in RPMI 1640 medium (Lonza, Walkersville, MD, USA) containing penicillin/streptomycin (Gibco, Grand Island, NY, USA) and 10% fetal bovine serum (FBS) (Atlas Biologicals, Fort Collins, CO, USA) in a humidified atmosphere with 5% CO_2_ at 37 °C. Cells were grown to an 80–90% confluency, and the medium was refreshed every 3–4 days [52].

#### 3.3.6. Anti-Inflammatory Activity of Digested Camel Milk Yogurt

The cells were seeded at 2 × 10^5^ cells/mL at 37 °C in 5% CO_2_ in McCoy’s 5A Medium (Gibco) with streptomycin–penicillin and 10% FBS. The media was refreshed every 2–3 days. The cells were subcultured in 48-well plates and washed twice with HBSS (Hank’s balanced salt solution). The HT-29 cells were pretreated with digested camel milk carao yogurts with FBS (5%) (1:25 *w*/*v*) (preliminary cell viability test) for 2 h and induced with the inflammatory factor of tumor necrosis factor-α (TNF-α) (2 ng∙mL^−1^) and for 4 h. The release of interleukin-8 (IL-8) was measured by collecting the supernatant of wells, and it was kept at −80 °C [35]. The IL-8 levels were estimated with an ELISA kit (eBioscience, Inc., San Diego, CA, USA) in a 96-well plate. The samples were incubated using PBS (phosphate-buffered saline) (100 µL) and mouse anti-human IL-8 antibodies. The supernatant of samples (100 µL) and standard were mixed and incubated for 2 h at 25 °C. Then, wells were washed and blocked with 200 µL of blocking buffer for 1 h. A secondary antibody was included, and after 1 h, TMB (100 µL) and avidin-horseradish peroxidase conjugate (Av-HRP) (100 µL) were used for color development. 50 µL 0.17 M H_3_PO_4_ was added in the final step to control the reaction, and absorbance was measured at 450 nm [53].

#### 3.3.7. Real-Time Reverse-Transcriptase Polymerase Chain Reaction Analysis

The relative gene expression was observed for tumor necrosis factor-α (TNF-α) (Primers, 5′-AAG CCC TGG TAT GAG CCC ATC TAT-3′ and 5′-AGG GCA ATG ATC CCA AAG TAG ACC-3′) and interleukin-1β (IL-1β) (Primers, 5′-TAC CTG AGC TCG CCA GTG AAA T-3′, and 5′-CCT GGA AGG AGC ACT TCA TCT GTT-3′). Cells were treated with the digested camel milk yogurt for 15 h and induced with lipopolysaccharide (LPS) (1 µg/mL) for 12 h. The HT-29 epithelial cells were seeded in 6-well plates, and total RNA was extracted using TRIzol reagent (Ambion, Austin, TX, USA). The TOPscript RT DryMIX kit (Enzynomics, Daejeon, Korea) was used to observe the reverse transcription. The relative gene expression was measured using the 2 × Real-Time PCR mix and Real-Time PCR System (Thermo Fisher Scientific, Pittsburgh, PA, USA). The thermal requirements were 97 °C for 17 min, ensued by 42 cycles of 97 °C for 25 s and 60 °C for 43 s, followed by 59 °C for 33 s and maintained at 3 °C. Relative gene expression was calibrated by the GAPDH gene (Primers, 5′-GAC CCC TTC ATT GAC CTC AAC TAC-3′, and 5′-ATG ACA AGC TTC CCG TTC TCA G-3′) [44].

#### 3.3.8. Sensory Evaluation

A completely randomized design (CRD) was used to investigate the effect of carao on the sensory properties of yogurt samples. This study was approved by the Honduran Association of Physicians-Nutritionists (ASOHMENU) with codige form # AS-ASHOMENU-013-2023. Consumers (*n* = 100) from UNAG (Catacamas, Olancho, Honduras) participated in the sensory analysis using partitioned sensory booths. Yogurt (30 mL plastic cup) was coded with a three-digit random number. Yogurt were tasted at a refrigerated temperature of 4 °C. The consumers (untrained) were previously instructed to evaluate the sensory attributes of camel milk yogurts (appearance, color, consistency, aroma, sweetness, sourness, aftertaste, flavor, smoothness, and overall liking) using a nine-point hedonic scale (1 = dislike extremely; 5 = neither like nor dislike; 9 = like extremely). Each sample’s purchase intent and acceptability were examined using a binomial scale (1 = Yes; 2 = No). The consumers were asked two questions without a beneficial statement (Before) and with a beneficial statement (After). The questions were as follows: “Would you purchase this product if it was available at a proper price?”, “ Would you purchase this product knowing that this product has higher antioxidant and anti-inflammatory properties than regular plain yogurt?”. According to their preference, the panelist was asked to rank all samples (1 = most preferred; 6 = least preferred) at the end of the questionnaire.

#### 3.3.9. Statistical Analysis

The data were analyzed using SPSS–PASW statistics software version 18.0 for Windows (SPSS, Chicago, IL, USA). The one-way ANOVA was used to analyze antioxidant capacity evaluations, anti-inflammatory activity, and sensory attributes. Proc mixed was used to determine the statistical significance of the carao concentration effect, time effect, and interaction effect between carao concentration and time on physico-chemical evaluations and microbial counts. The Bonferroni post hoc test was applied in the ANOVA and Proc mixed. Multiple pairwise comparisons were conducted using Cochran’s Q test for the purchase intent among carao yogurts (0, 1.3, 2.65, and 5.3 g/L). For the preference and ranking data, a Friedman analysis was conducted. The McNemar test was used to determine statistical differences in purchase intent before and after the beneficial statement was provided. A probit regression was used to determine the rejection tolerance threshold and range. *p*-values less than 0.05 was used to detect statistically significant differences.

## 4. Conclusions

Camel milk inoculated with microorganisms fermented despite the addition of carao powder. The results showed that camel milk yogurt with carao addition at different concentrations showed physico-chemical improvements, antioxidant activity, inflammatory effects, and prohibition effects in inflammatory mRNA. Thus, this camel milk yogurt elaborated could be an innovative product with high added value due to its high biological properties that could be marked. Sensory analysis of carao yogurts during extended storage should be evaluated to comprehend the overall changes in perception. Yogurt with other probiotics, such as *L. plantarum* or *L. casei*, should be studied to examine the prebiotic effect of carao.

## Figures and Tables

**Figure 1 pharmaceuticals-16-01032-f001:**
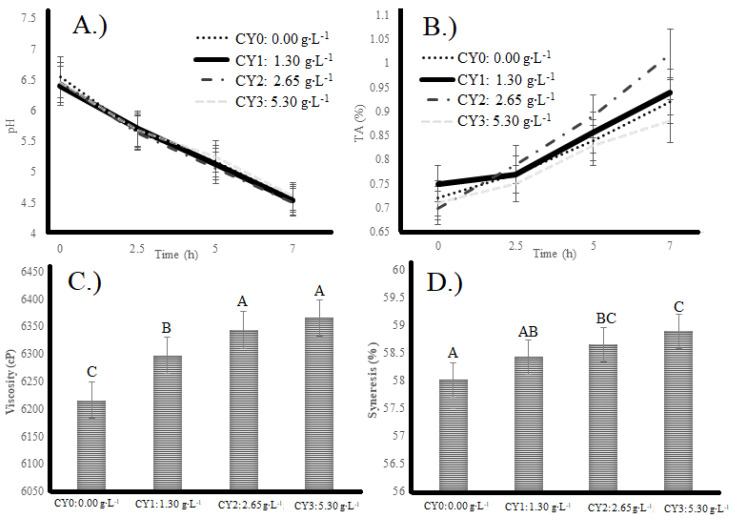
Physico-chemical attributes (**A**) pH, (**B**) titratable acidity, (**C**) viscosity, and (**D**) synerisis of camel milk yogurt with carao addition at different concentration. CY = Carao Yogurt. The letters show statistically significant differences between variables.

**Figure 2 pharmaceuticals-16-01032-f002:**
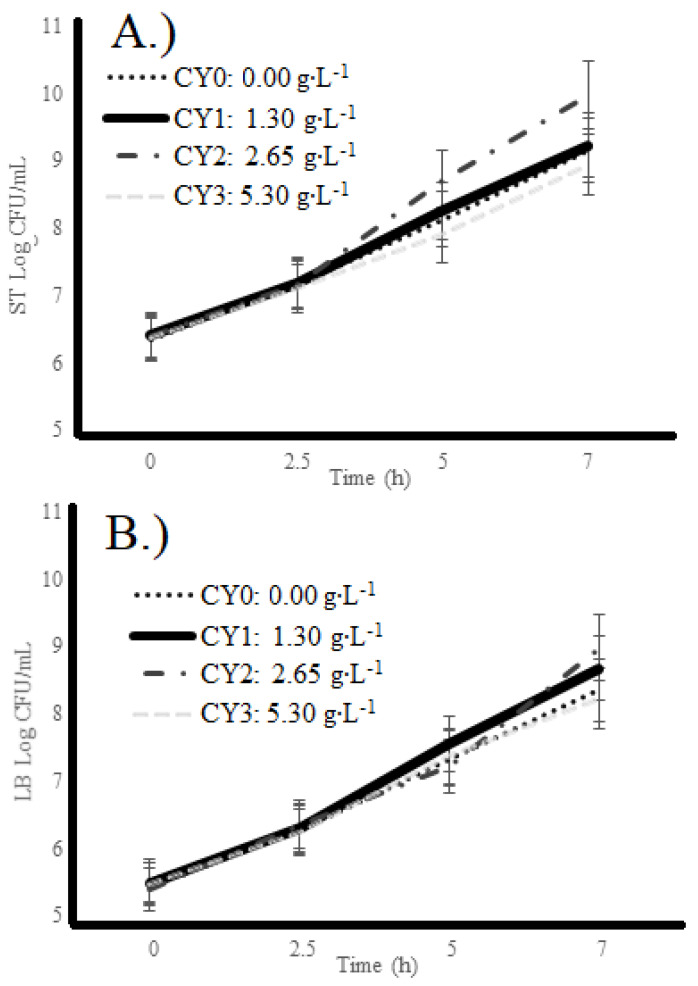
Microbial counts (Log CFU∙mL^−1^) of (**A**) *S. thermophilus* and (**B**) *L. bulgaricus* in camel milk yogurt manufactured with carao. CY0: 0.00 g∙L^−1^; CY1: 1.30 g∙L^−1^; CY2: 2.65 g∙L^−1^; and CY3: 5.30 g∙L^−1^; CY = Carao Yogurt.

**Figure 3 pharmaceuticals-16-01032-f003:**
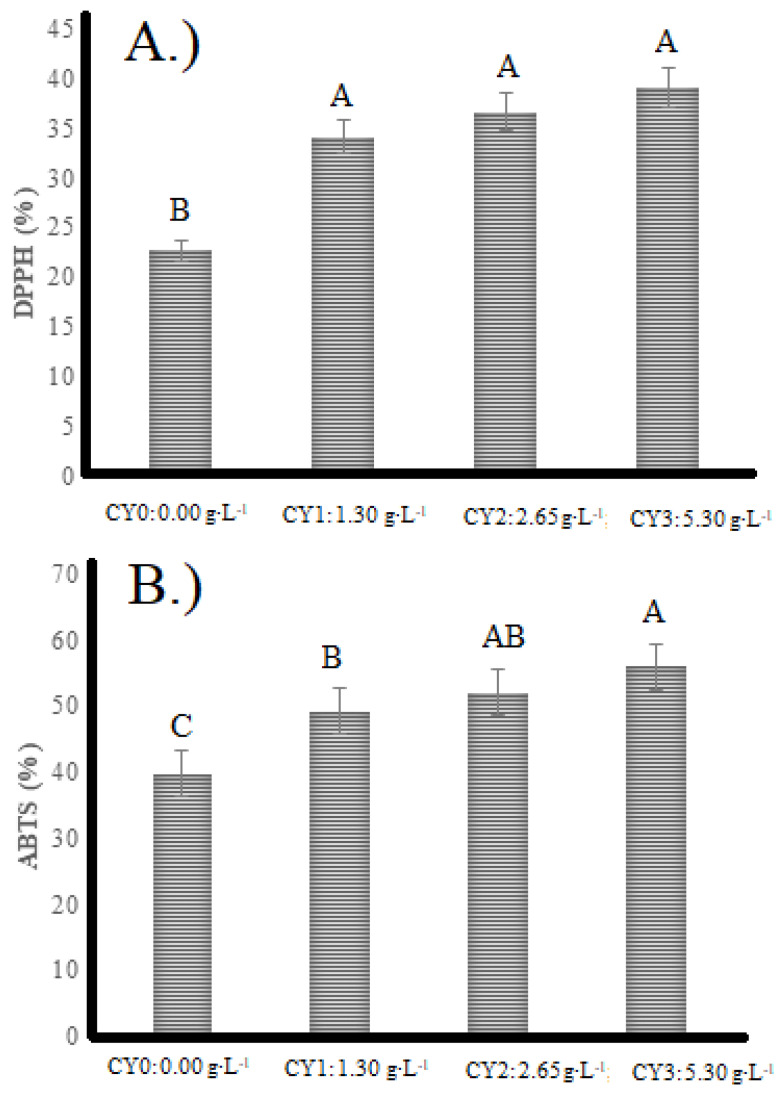
Antioxidant capacity of camel milk yogurts. CY = Carao Yogurt. Results of the radical scavenging ability of DPPH (**A**) and ABTS (**B**). The letters show statistically significant differences between variables.

**Figure 4 pharmaceuticals-16-01032-f004:**
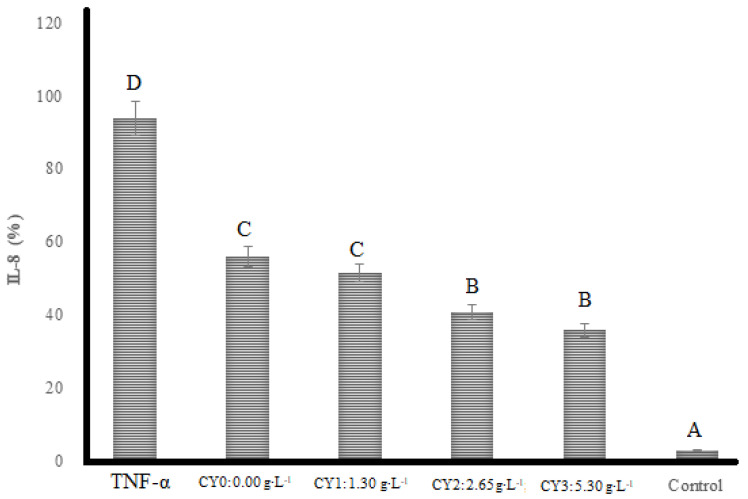
Interleukin 8 (IL-8) (%) production responses to TNF-inflammatory factor of camel milk yogurt in HT-29 cells. TNF-α: Tumor necrosis factor-α; Carao yogurt (CY), and Control: without TNF-α and yogurt addition. The letters show statistically significant differences between variables.

**Figure 5 pharmaceuticals-16-01032-f005:**
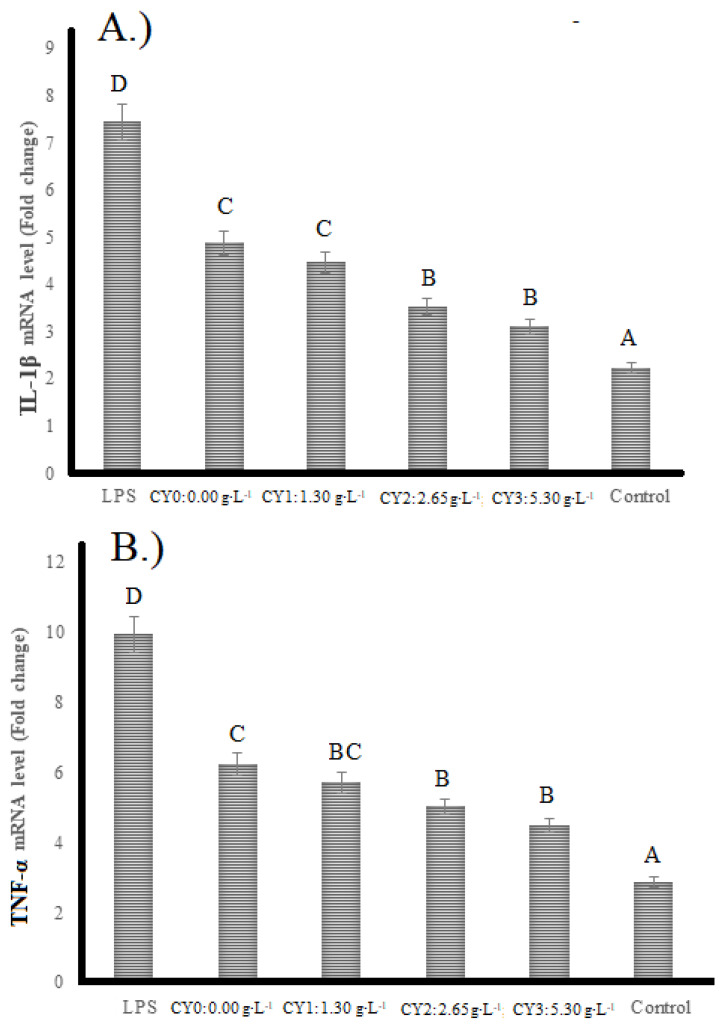
Real-time PCR mRNA expression of interleukin-1β (IL-1β) (**A**) and TNF-α (**B**) in HT-29 cells treated with carao yogurts. LPS: lipopolysaccharide; CY: Carao yogurt, and Control: without LPS and carao addition. The letters show statistically significant differences between variables.

**Figure 6 pharmaceuticals-16-01032-f006:**
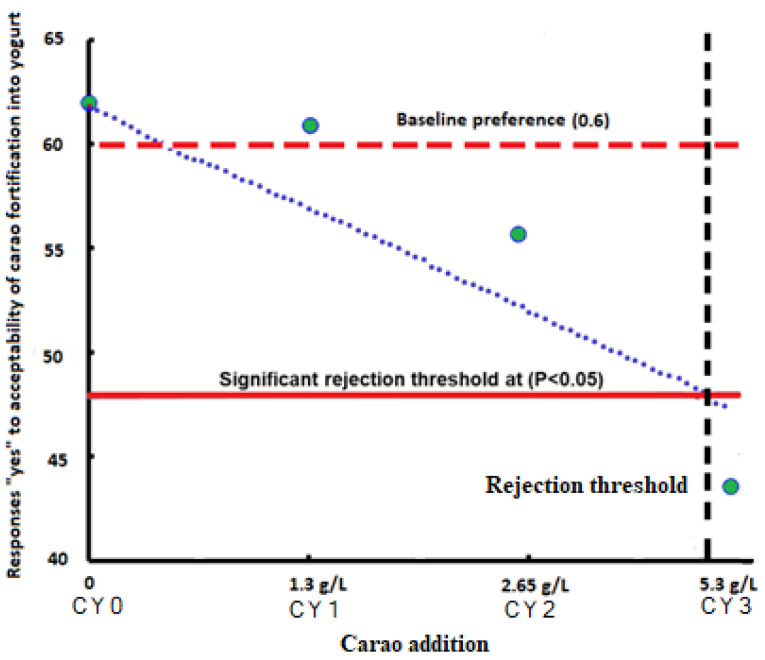
Sensory responses acceptability of tasters of camel milk yogurt with different concentration of carao powder. Black dashed line refers to the rejection threshold expected to result in 90% overall acceptability (equivalent to a 10% rejection tolerance) of yogurt. CY0: 0.00 g∙L^−1^; CY1: 1.30 g∙L^−1^; CY2: 2.65 g∙L^−1^ and CY3: 5.30 g∙L^−1^ carao.

**Table 1 pharmaceuticals-16-01032-t001:** Sensory evaluation of camel milk yogurt with different concentration of carao powder. Different lowercase letters mean statistically significant differences between the different treatments for each attribute (one-way ANOVA followed by Tukey’s test, *p* < 0.05).

Formulations	Appearance	Color	Consistency	Aroma	Sweetness
CY0	6.43 ± 1.54 a	6.57 ± 1.602	6.51 ± 1.91 a	6.38 ± 1.80 a	6.19 ± 1.54 a
CY1	6.34 ± 1.33 a	6.21 ± 1.55 a	6.28 ± 1.35 ab	6.02 ± 1.26 a	6.02 ± 1.88 a
CY2	6.25 ± 1.38 ab	6.06 ± 2.11 b	6.06 ± 1.89 b	5.11 ± 1.58 b	6.07 ± 1.21 a
CY3	6.01 ± 1.74 b	6.04 ± 2.12 b	6.03 ± 1.79 b	5.28 ± 1.85 b	6.41 ± 1.94 a
**Formulations**	**Sourness**	**After-Taste**	**Thickness**	**Flavor**	**Overall Liking**
CY0	6.30 ± 1.25 a	6.44 ± 1.77 a	6.15 ± 1.47 a	6.77 ± 1.30 a	6.68 ± 1.49 a
CY1	6.13 ± 1.96 a	6.45 ± 1.38 a	6.17 ± 1.24 a	6.66 ± 1.68 a	6.49 ± 1.16 a
CY2	6.11 ± 1.87 a	6.26 ± 1.37 a	6.42 ± 1.73 a	6.05 ± 1.27 b	5.92 ± 1.79 b
CY3	6.34 ± 1.79 a	6.15 ± 1.48 a	6.17 ± 1.90 a	6.15 ± 1.94 b	5.76 ± 1.72 b

CY0: 0.00 g∙L^−1^; CY1: 1.30 g∙L^−1^; CY2: 2.65 g∙L^−1^; and CY3: 5.30 g∙L^−1^; CY = Carao Yogurt.

**Table 2 pharmaceuticals-16-01032-t002:** Purchase intent (%) and ranking of camel milk yogurt with different concentration of carao powder using a binomial scale (1 = Yes, 2 = No).

Formulations	Purchase Intent (PI, %)		Ranking (Rank Sums)
	PI-Before	PI-After	Preference
CY0	68.23% ^a,A^	-	160 ^a^
CY1	59.54% ^a,A^	71.45% ^a,B^	165 ^a,b^
CY2	61.56% ^a,A^	68.56% ^a,B^	183 ^b^
CY3	55.67% ^b,A^	67.32% ^a,B^	180 ^b^

^a^ Same superscripts within the same column were not significantly different (*p* ≥ 0.05; Cochran Q test) in purchase intent. ^A–B^, the same letter within the same row were not significantly different (*p* ≥ 0.05; McNemar test) in purchase intent. (^a–b^) the same column were not significantly different (*p* ≥ 0.05; Friedman test) in the rank sums. CY0: 0.00 g∙L^−1^; CY1: 1.30 g∙L^−1^; CY2: 2.65 g∙L^−1^; and CY3: 5.30 g∙L^−1^; CY = Carao Yogurt.

## Data Availability

The authors confirm that the data supporting the findings of this study are available within the article and the raw data that support the findings are available from the corresponding author, upon reasonable request.

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
