# Peer review of "Isolated Fraction of Gastric-Digested Camel Milk Yogurt with Carao (*Cassia grandis*) Pulp Fortification Enhances the Anti-Inflammatory Properties of HT-29 Human Intestinal Epithelial Cells"

_pharmaceuticals, 2023, doi:10.3390/ph16071032_

Round 1

Reviewer 1 Report

The article in general presents a high scientific soundness. This article is good structured,  the introduction provides sufficient background and includes all relevant references

Minor editing of English language require

Author Response

Reviewer # 1

Comments and Suggestions for Authors

The article in general presents a high scientific soundness. This article is good structured,  the introduction provides sufficient background and includes all relevant references

Comments on the Quality of English Language

Minor editing of English language require

Response: English in the manuscript was addressed.

Reviewer 2 Report

The manuscript presents the results of fortification of Camel milk yogurt with Carao regarding fermentation characteristics, anti-inflammatory and oxidative properties, as well as sensory parameters. The theme is original and presents contributions to produce healthier products in the market. However, important controls are lacking, particularly regarding to digestion and anti-inflammatory analyses. I indicate the publication of this manuscript only when these issues are resolved.

Line 31: specify the abbreviation TA

Section 2.1: it is not clear if the authors use the whole part of the plant (pulp, seeds, and shells) or just one of them.

Section 2.3.3

the authors cannot say that they performed a simulated digestion, considering that they did not perform the oral or intestinal phase. The model used only performs a hydrolysis with pepsin. Therefore I suggest  to change “simulated gastric digestion” expression.

the authors only considered protein digestion. Neither lipid nor carbohydrate digestion was considered in this analysis. Besides, there is no information regarding the protein content of the sample.

specify the final yogurt dilution volume. Initially, 0.2 g in 4 mL of HCl were added, however the volume of gastric solution is not mentioned.

It is recommended that the authors evaluate the bioactivity of digestion control (replace the sample with water), in order to assess whether the digestate itself does not present bioactivity on the cells. It is necessary to considerer that the samples were not ultrafiltered, and that pepsin is probably contaminating the samples.

it is strongly recommended to characterize the sample in relation to protein, carbohydrates, fat, salts, total phenolic content, in order to better relate it to the bioactivity observed in vitro.

Line 332: Fig 4 shows the content of IL8 secreted and not only produced.

Section 3.4 and 3.5:

the authors do not perform a cell viability test with the treatments, neither sample or stimulus. It is not possible to state that the treatment is not compromising cell viability, therefore, it is not possible to state that the results are via activation of anti-inflammatory pathway or anti-apoptotic route.

The authors should explain why they differ in inflammatory stimulus (TNFa and LPS) between IL8 secretion and mRNA test

Why they chose different times of treatment?

Line 352: specify which receptor is referring to. TLR-4?

Line 358: “Less IL-1β secretion was observed in the sample without LPS induction and Carao addition.” Authors did not observe secretion, but production of mRNA.

Pay attention on describing TNFa and IL1b with out the mRNA in front of the word. can be a confounder regarding levels of the protein perse.

Author Response

Reviewer # 2

Comments and Suggestions for Authors

The manuscript presents the results of fortification of Camel milk yogurt with Carao regarding fermentation characteristics, anti-inflammatory and oxidative properties, as well as sensory parameters. The theme is original and presents contributions to produce healthier products in the market. However, important controls are lacking, particularly regarding to digestion and anti-inflammatory analyses. I indicate the publication of this manuscript only when these issues are resolved.

Line 31: specify the abbreviation TA

Response: abbreviation TA was define as (titratable acidity).

Section 2.1: it is not clear if the authors use the whole part of the plant (pulp, seeds, and shells) or just one of them.

Response: Just he pulp was used so the title was modified and plant material was described appropriately.

Section 2.3.3

the authors cannot say that they performed a simulated digestion, considering that they did not perform the oral or intestinal phase. The model used only performs a hydrolysis with pepsin. Therefore I suggest  to change “simulated gastric digestion” expression.

Response: the “simulated gastric digestion” expression was changed to simulated gastric phase digestion.

the authors only considered protein digestion. Neither lipid nor carbohydrate digestion was considered in this analysis. Besides, there is no information regarding the protein content of the sample.

Response: Low fat yogurt is best known for anti-inflammatory properties (Zhai et al., 2019) when considering invitro studies in intestinal cells. Also, Yogurt peptides are known to have inflammatory properties (Putt et al., 2017). As a result, the focus was on protein digestion since most the of the biological value of yogurt is on its protein content with its probiotic content.

Zhai, Zhengyuan, Jiaojiao Wang, Baozhu Huang, and Sheng Yin. "Low-fat yogurt alleviates the pro-inflammatory cytokine IL-1β-induced intestinal epithelial barrier dysfunction." Journal of dairy science 102, no. 2 (2019): 976-984.

Putt, Kelley K., Ruisong Pei, Heather M. White, and Bradley W. Bolling. "Yogurt inhibits intestinal barrier dysfunction in Caco-2 cells by increasing tight junctions." Food & function 8, no. 1 (2017): 406-414.

specify the final yogurt dilution volume. Initially, 0.2 g in 4 mL of HCl were added, however the volume of gastric solution is not mentioned.

Response: the concentrations of each component was better specified As follows “Yogurts were freeze-dried, and 0.2 g of powder was dissolved in 4 mL of HCl (0.15 N) and then mixed with simulated gastric fluid (Chemazone, Edmonton, Alberta, Canada) and porcine pepsin enzyme (Sigma-Aldrich, St. Louis, MO, USA) solution (Mix 1:1) at a 1:50 ratio (w/w) with sample solution (yogurt with HCL)”

It is recommended that the authors evaluate the bioactivity of digestion control (replace the sample with water), in order to assess whether the digestate itself does not present bioactivity on the cells. It is necessary to considerer that the samples were not ultrafiltered, and that pepsin is probably contaminating the samples.

Response: the focus of our study of the evaluate if the addition of carao improves anti-inflammatory properties of GHt29 cells so our control considered was just plain yogurt without carao. Chen et al., (2019) test the efficacy of plant-based yogurt in HT-29 cells so we focused in the carao fortification. As a result our control is just yogurt without carao.

Chen, Yuhuan, Hua Zhang, Ronghua Liu, Lili Mats, Honghui Zhu, K. Peter Pauls, Zeyuan Deng, and Rong Tsao. "Antioxidant and anti-inflammatory polyphenols and peptides of common bean (Phaseolus vulga L.) milk and yogurt in Caco-2 and HT-29 cell models." Journal of Functional Foods 53 (2019): 125-135.

it is strongly recommended to characterize the sample in relation to protein, carbohydrates, fat, salts, total phenolic content, in order to better relate it to the bioactivity observed in vitro.

Response: We acknowledge that this would make the manuscript stronger but due to the fact that there is no more funding for this project this data is missing. Nevertheless, this manuscript reveals that carao fortification improves the bio-viability of HT 29 cells. For future studies, we encourage to characterize the sample to try to explain what is causing the improvements observed.

Line 332: Fig 4 shows the content of IL8 secreted and not only produced.

Response: Fig 4 shows the content Interleukin 8 (IL-8) (%) production responses to TNF- inflammatory factor of Camel milk yogurt in HT-29 cells.

Section 3.4 and 3.5:

the authors do not perform a cell viability test with the treatments, neither sample or stimulus. It is not possible to state that the treatment is not compromising cell viability, therefore, it is not possible to state that the results are via activation of anti-inflammatory pathway or anti-apoptotic route.

Response: In our preliminary testing’s, the best concentration of yogurt/water dilution was determined. Thank you for bring this out. The preliminary cell viability testing was stated in the methods section.

The authors should explain why they differ in inflammatory stimulus (TNFa and LPS) between IL8 secretion and mRNA test

Response: We explained why we differ due to the amount of resources available to try to make the best experiments as possible. In the future, it will be good to study more cytokines in IL8 secretion and mRNA abundance.

Why they chose different times of treatment?

 Response: The methodology of Chen et al., (2019) was followed.

Chen, Yuhuan, Hua Zhang, Ronghua Liu, Lili Mats, Honghui Zhu, K. Peter Pauls, Zeyuan Deng, and Rong Tsao. "Antioxidant and anti-inflammatory polyphenols and peptides of common bean (Phaseolus vulga L.) milk and yogurt in Caco-2 and HT-29 cell models." Journal of Functional Foods 53 (2019): 125-135.

Line 352: specify which receptor is referring to. TLR-4?

Response: Yes, TLR-4 was specified on line 352.

Line 358: “Less IL-1β secretion was observed in the sample without LPS induction and Carao addition.” Authors did not observe secretion, but production of mRNA.

Response: thank you for this comment. The sentence was changed to “Less mRNA levels of IL-1β was observed in the sample without LPS induction and Carao addition.”

Pay attention on describing TNFa and IL1b with out the mRNA in front of the word. can be a confounder regarding levels of the protein perse.

Response: thank you for this comment.

Reviewer 3 Report

Manuscript Number: pharmaceuticals-2445128

Title: Camel milk yogurt with Carao (Cassia grandis) fortification enhances the antioxidant and anti-inflammatory properties of HT-29 human intestinal epithelial cells

The manuscript entitledCamel milk yogurt with Carao (Cassia grandis) fortification enhances the antioxidant and anti-inflammatory properties of HT-29 human intestinal epithelial cells» showed that Camel milk yogurt with Carao addition at different con- 429 centrations showed physicochemical improvements, antioxidant activity, inflammatory 430 effects, and prohibition effects in inflammatory mRNA.

Although some of the results are interesting, the manuscript is not well writing, the objectives are not clear. Some of the results are surprising and more experiments are required to clearly demonstrate some of the main conclusions raised in this manuscript. The discussion is not appropriated y basically repeats the results.

English very difficult to understand/incomprehensible

Author Response

Reviewer # 3

Comments and Suggestions for Authors

Title: Camel milk yogurt with Carao (Cassia grandis) fortification enhances the antioxidant and anti-inflammatory properties of HT-29 human intestinal epithelial cells

The manuscript entitled “Camel milk yogurt with Carao (Cassia grandis) fortification enhances the antioxidant and anti-inflammatory properties of HT-29 human intestinal epithelial cells» showed that Camel milk yogurt with Carao addition at different con- 429 centrations showed physicochemical improvements, antioxidant activity, inflammatory 430 effects, and prohibition effects in inflammatory mRNA.

Although some of the results are interesting, the manuscript is not well writing, the objectives are not clear. Some of the results are surprising and more experiments are required to clearly demonstrate some of the main conclusions raised in this manuscript. The discussion is not appropriated y basically repeats the results.

Response: Thank you for the feedback the  discussion, conclusion and English was improved throughout the manuscript.

Reviewer 4 Report

Camel Milk Yogurt with Carao Fortification Enhances Antioxidant and Anti-Inflammatory Properties of HT-29 Human Intestinal Epithelial Cells

Abstract:

The abstract provides a concise summary of the research study, which focuses on examining the potential benefits of fortifying camel milk yogurt with carao (Cassia grandis) fruit. The study aims to evaluate the impact of carao incorporation on anti-inflammatory and antioxidant properties, as well as the physiochemical and sensory attributes of camel milk yogurt. The findings indicate that the addition of carao to yogurt enhances various parameters, including antioxidant capacity, viscosity, syneresis, and bacterial counts, while reducing the inflammatory response.

Introduction:

The introduction provides a clear context for the study by highlighting the increasing popularity of functional foods and the interest in using yogurt as a carrier for bioactive compounds. The rationale for investigating carao and camel milk as potential sources of functional ingredients is well-established. However, the introduction could benefit from providing more background information on the anti-inflammatory and antioxidant properties of carao and its potential health benefits. Additionally, a more detailed explanation of the relevance of HT-29 cells as a model for anti-inflammatory response would enhance the understanding of the study.

Methodology:

The methodology section provides a general overview of the experimental design and procedures used in the study. However, it lacks sufficient detail to allow for replication or evaluation of the methodology's robustness. Key information, such as the sample size, experimental controls, and statistical analyses employed, should be included. Furthermore, more information about the source of carao, camel milk, and HT-29 cells, as well as the extraction and fortification processes, would be valuable for better understanding the study.

Results and Discussion:

The results and discussion section presents the key findings of the study. The data indicate that the inclusion of carao in camel milk yogurt leads to significant improvements in several parameters, such as pH, titratable acidity, viscosity, syneresis, antioxidant capacity, and bacterial counts. Moreover, the anti-inflammatory response, as measured by IL-8 and mRNA production of IL-1β and TNF-α, is significantly reduced with the addition of carao. However, the section lacks a comprehensive interpretation of the results, including the underlying mechanisms or potential limitations of the study. Additionally, a comparison with previous studies or relevant literature would strengthen the discussion.

Sensory Evaluation and Consumer Study:

The study incorporates a sensory evaluation and consumer study to assess the impact of carao addition on the sensory properties of the yogurt. However, the report does not provide sufficient details regarding the methods employed for sensory evaluation, such as the selection of panelists, sensory attributes evaluated, and the statistical analysis of sensory data. These details are crucial for evaluating the reliability and significance of the sensory findings.

Conclusion:

The conclusion provides a brief summary of the study's findings. However, it does not offer a comprehensive discussion of the implications, limitations, or potential applications of the research. The report would benefit from a more detailed conclusion that highlights the significance of the findings, addresses any potential shortcomings, and suggests directions for future research.

Overall Assessment:

Overall, the study presents promising findings regarding the enhanced antioxidant and anti-inflammatory properties of camel milk yogurt fortified with carao. Figures 1-3 need some improvement, as data is not understandable. However, several areas need improvement, such as providing more background information, expanding the methodology section, interpreting the results in greater detail, and offering a more comprehensive conclusion. Addressing these limitations would enhance the overall quality and impact of the research.  I would suggest publication of this paper after minor revision.

Author Response

Comments have been included in the manuscript

Reviewer 5 Report

Manuscript: pharmaceuticals-2445128

Camel milk yogurt with Carao (Cassia grandis) fortification enhances the antioxidant and anti-inflammatory properties of HT-29 human intestinal epithelial cells

General comments:

This manuscript reports findings on development of functional camel milk yogurt incorporating different concentration of Carao (0, 1.3, 2.65, and 5.3 g/L). Physiochemical, sensory properties, anti-inflammatory stimulus and antioxidant activity and of the product were evaluated. This study concludes that camel milk yogurt with Carao addition at different concentrations showed physicochemical improvements, antioxidant activity, inflammatory effects, and prohibition effects in inflammatory mRNA. Thus, this Camel milk yogurt elaborated could be an innovative product with high added value due to its high biological properties that could be marked. In general, this study covers a number of useful characterization techniques to evaluate the research hypothesis.

In my opinion the topic and research subject are interesting and has enough novelty enabling this manuscript to be published in Pharmaceuticals. However, I would suggest flowing points to be considered by authors to improves the quality of this manuscript.

 Abstract:

1)      Abstract has been well-written however followings can be added to improve its quality:

·                hypothesis (aim) of the study (after short introduction)

·                most important results (quantitative) statistically-supported

·                a general conclusion,

·                future prospective (implication of the results for industrialization of this product).

Introduction:

2)      Please give a brief survey of literature about similar reports on application of Carao in dairy products and explain the novelty of your research compered to those literature.

3)      Please provide the “research hypothesis” at the end of your introduction.

Materials & Methods:

Well-written.

 Results and Discussion:

Fine.

Author Response

(The authors gave the same response as above.)

Round 2

Reviewer 2 Report

The authors' responses were not enough to improve the manuscript and maintain the journal's publication quality. Additional experiments are needed. Some answers were not answered properly as follows:

Second review

the authors only considered protein digestion. Neither lipid nor carbohydrate digestion was considered in this analysis. Besides, there is no information regarding the protein content of the sample.

Response: Low fat yogurt is best known for anti-inflammatory properties (Zhai et al., 2019) when considering invitro studies in intestinal cells. Also, Yogurt peptides are known to have inflammatory properties (Putt et al., 2017). As a result, the focus was on protein digestion since most the of the biological value of yogurt is on its protein content with its probiotic content.

Response 2: although the bioactive properties are mostly from protein hydrolysis, it is important to characterize the sample and identify at least the protein percentage. Carbohydrates and lipids can interact with other components, interfering in the analysis. Don’t you have the data of protein content of your sample?

It is recommended that the authors evaluate the bioactivity of digestion control (replace the sample with water), in order to assess whether the digestate itself does not present bioactivity on the cells. It is necessary to considerer that the samples were not ultrafiltered, and that pepsin is probably contaminating the samples.

Response: the focus of our study of the evaluate if the addition of carao improves anti-inflammatory properties of GHt29 cells so our control considered was just plain yogurt without carao. Chen et al., (2019) test the efficacy of plant-based yogurt in HT-29 cells so we focused in the carao fortification. As a result our control is just yogurt without carao.

Response 2: Please, specify in methods if your control with yogurt without carao pass through the same gastric digestion process than carao sample.

it is strongly recommended to characterize the sample in relation to protein, carbohydrates, fat, salts, total phenolic content, in order to better relate it to the bioactivity observed in vitro.

Response: We acknowledge that this would make the manuscript stronger but due to the fact that there is no more funding for this project this data is missing. Nevertheless, this manuscript reveals that carao fortification improves the bio-viability of HT 29 cells. For future studies, we encourage to characterize the sample to try to explain what is causing the improvements observed.

Response 2: Your evaluated is all based in peptide bioactivity. At least protein content (percentage of total sample) is necessary.

the authors do not perform a cell viability test with the treatments, neither sample or stimulus. It is not possible to state that the treatment is not compromising cell viability, therefore, it is not possible to state that the results are via activation of anti-inflammatory pathway or anti-apoptotic route.

Response: In our preliminary testing’s, the best concentration of yogurt/water dilution was determined. Thank you for bring this out. The preliminary cell viability testing was stated in the methods section.

Response 2: If you tested cell viability you must show the results. Which assay was evaluated? This is important data for interpretation of results.

The authors should explain why they differ in inflammatory stimulus (TNFa and LPS) between IL8 secretion and mRNA test

Response: We explained why we differ due to the amount of resources available to try to make the best experiments as possible. In the future, it will be good to study more cytokines in IL8 secretion and mRNA abundance.

Response 2: I did not understand the relation of amount of resources with different stimuli. The use of a single stimulus could even save time and money, since the extracellular content for IL-8 and the mRNA analyzes could have been done in the same culture dish.

Why they chose different times of treatment?

Response: The methodology of Chen et al., (2019) was followed.

Chen, Yuhuan, Hua Zhang, Ronghua Liu, Lili Mats, Honghui Zhu, K. Peter Pauls, Zeyuan Deng, and Rong Tsao. "Antioxidant and anti-inflammatory polyphenols and peptides of common bean (Phaseolus vulga L.) milk and yogurt in Caco-2 and HT-29 cell models." Journal of Functional Foods 53 (2019): 125-135.

Response 2: Chen, 2019 used 4h hour treatment, and the authors in this manuscript used 4 h for IL-8 secretion and 27 h for mRNA expression. This leads me to suspect that the authors chose the best analysis time, omitting intermediate times.

Author Response

Dear editor, the corrected article is sent

Reviewer 3 Report

no comments

Author Response

no comments